# Influence of In-Situ Electrochemical Oxidation on Implant Surface and Colonizing Microorganisms Evaluated by Scanning Electron Microscopy

**DOI:** 10.3390/ma12233977

**Published:** 2019-11-30

**Authors:** Maximilian Göltz, Maximilian Koch, Rainer Detsch, Matthias Karl, Andreas Burkovski, Stefan Rosiwal

**Affiliations:** 1Division of Ultra-Hard Coatings, Department of Material Sciences, University of Erlangen-Nuremberg, 91058 Erlangen, Germany; maximilian.goeltz@fau.de (M.G.); Stefan.Rosiwal@fau.de (S.R.); 2Microbiology Division, Department of Biology, University of Erlangen-Nuremberg, 91058 Erlangen, Germany; Maximilian.G.F.Koch@gmx.de (M.K.); andreas.burkovski@fau.de (A.B.); 3Institute of Biomaterials, Department of Material Sciences, University of Erlangen-Nuremberg, 91058 Erlangen, Germany; rainer.detsch@fau.de; 4Department of Prosthodontics, Saarland University, 66424 Homburg/Saar, Germany

**Keywords:** *Candida*, *Enterococcus*, debridement, peri-implantitis, disinfection, microscopy, electron, scanning

## Abstract

Peri-implantitis is a worldwide increasing health problem, caused by infection of tissue and bone around an implant by biofilm-forming microorganisms. Effects of peri-implantitis treatment using mechanical debridement, air particle abrasion and electrochemical disinfection on implant surface integrity were compared. Dental implants covered with bacterial biofilm were cleaned using mechanical debridement and air particle abrasion. In addition, implants were disinfected using a novel electrochemical technique based on an array of boron-doped diamond (BDD) coated electrodes. Following treatment and preparation, the implants were inspected by scanning electron microscopy (SEM) and energy dispersive X-ray spectroscopy (EDX). Mechanical debridement led to changes in surface topography destroying the manufacturer’s medium-rough surface by scratch formation. Air particle abrasion led to accumulation of the abrasive used on the implant surface. With both treatment options, appearance of bacteria and yeasts was not affected. In contrast, electrochemical disinfection did not cause alterations of the implant surface but resulted in distorted microbial cells. Electrochemical disinfection of implant surfaces using BDD electrodes may constitute a promising treatment option for cleaning dental implant surfaces without negatively affecting materials and surface properties.

## 1. Introduction

Despite lacking a proper definition, peri-implantitis is understood as constituting an inflammatory, plaque-induced disease of the tissue surrounding an implant, which may lead to progressive bone loss compromising implant survival [1]. Besides patient-related factors such as preceding periodontitis and general health, restorative factors such as the design of the superstructure and hygiene efforts are currently understood as playing a role in the multifactorial pathogenesis of peri-implantitis [2]. In contrast to restorative materials [3] it has so far not been possible to create implant surfaces [4] which are conducive to osseointegration but avoid bacterial colonization. A plethora of non-surgical [5] and surgical [6,7] treatment protocols have been established indicating that an optimal strategy with successful and predictable outcomes does not yet exist. Techniques for dental implant decontamination used alone or in combination include mechanical debridement [8], use of antimicrobial agents including the local or systemic application of antibiotics [9,10,11,12], laser application [13,14], photodynamic therapy [15,16,17], cold plasma treatment [18,19] and air particle abrasion [20,21,22,23] (for review, see [24]).

Some of the established techniques may impair the surface topography or alter the surface properties in a way that bacterial recolonization is favored or re-osseointegration is obstructed [14,25,26,27]. Consequently, there is a need for the development of new and reliable methods lacking undesirable side effects. Recent examples for such approaches are the successful application of an Er, Cr:YSGG laser to decontaminate artificially infected implants [14] and the electrochemical removal of biofilm from dental implant surfaces using the implant itself as an electrode [28].

In this study, a recent concept for the in-situ electrochemical oxidation using boron-doped diamond (BDD) electrodes [29] was tested. During a so-called advanced oxidation process (AOP) [30] the high anodic overpotential of BDD against O_2_ formation (approximately 2.5 V) generates different reactive oxidation species such as hydroxyl radicals and ozone directly at the diamond surface of the electrode (for standard redox potentials see Table 1).

Recently, BDD electrodes have been described as being suitable for large-scale electrochemical disinfection of microbially contaminated water [31] and we were already able to show that also root canals and implants can be disinfected with low electric current ranging from 2.5 to 9 V, which is considered as being harmless for the patient [32,33].

It was the aim of this proof-of-principle study to investigate potential alterations of implant surface topography following BDD treatment. In the frame of this communication, we compared mechanical debridement by curettes, air particle abrasion and electrochemical disinfection of implants artificially contaminated by microorganisms, assuming that none of the treatments would cause surface alterations of the implants. As colonizing species, the yeast *Candida dubliniensis* and the Gram-positive bacterium *Enterococcus faecalis* were chosen. These pathogens were isolated from infected root canals by us recently [34,35], are frequently found in cases of peri-implantitis [36,37] and are in contrast to the anaerobic or microaerophilic members of the genera *Prevotella*, *Porphyromonas* and *Treponema*, which are highly oxygen-sensitive, fast growing and resistant against atmospheric oxygen concentrations.

## 2. Materials and Methods

### 2.1. Preparation of Contaminated Implant Surfaces

Dental implants (Figure 1) with a medium-rough surface obtained through sandblasting and acid etching (Straumann Bone Level Tapered 4.1 × 12 mm RC; REF: 021.5512; LOT: RP027) were incubated in rich medium (Brain Heart Infusion (BHI); Oxoid, Wesel, Germany), and inoculated with *C. dubliniensis* and *E. faecalis*, respectively, at 37 °C for three days to allow for biofilm formation.

### 2.2. Set-up of the Boron-Doped Diamond Electrode

Niobium wires (200 µm diameter, 99.9% Nb-containing, Nb-502, Haines & Maassen Metallhandelsgesellschaft mbH, Bonn, Germany) were pre-treated with sandblasting (5 bar, Cemat NT4, Wassermann Dental-Maschinen GmbH, Hamburg, Germany) using silicon carbide particles (17-74 µm, SiC F320) for optimal properties with respect to surface roughness for diamond adhesion and stiffness [38]. Sandblasted wires were cleaned in an ultrasonic bath (2 min, 45 kHz, Fa. Elma, Elmasonic X-tra) and diamond was seeded on the surface using nanodiamond dispersions (1:1000 or 1:10,000 in ethanol, Andante, Carbodeon Ltd. Oy, Vantaa, Finland). Diamond coating with boron doping was performed in a Hot-Filament Chemical Vapor Deposition machine (HFCVD). Tungsten filaments (220 mm, Ø 100 µm) were fit into a filament mount and pre-heated for carburization (18 h, 65 A/mount) in order to reach stable conditions during the deposition process following the pre-heating process in a methane-hydrogen-trimethyl borate gas atmosphere. Two pieces of wire, as electrodes, were combined with electrical insulating media to form a probe-like instrument with clinically applicable dimensions [32]. This probe was connected to an external electric power supply allowing for adjusting voltage and treatment time.

### 2.3. Decontamination of Implants

The implants colonized by microorganisms were rinsed with phosphate-buffered saline (137 mM NaCl, 2.7 mM KCl, 10 mM Na_2_HPO_4_, 2 mM KH_2_PO_4_, pH 7.4) and placed in elastic silicone, stable against temperature, acid, base and oxidants and bacterial colonization (Bindulin, Fürth, Germany). Subsequently, accessible implant surfaces were chemo-mechanically debrided using metal curettes or air particle abrasion (AIRFLOW PLUS, EMS ElectroMedicalSystems GmbH, Munich, Germany) in combination with chlorhexidine irrigation (Chlorhexamed FORTE ethanol-free 0.2%, GlaxoSmithKline Consumer Healthcare GmbH & Co. KG, Munich, Germany) as control [39]. In addition, a set of contaminated implants was treated using a BDD electrode. Samples were either used for microscopy or growth analyses [33]. Tests of growth behavior of bacteria and yeasts were highly reproducible with at least three independent biological replicates per species and treatment method applied.

### 2.4. Sample Preparation for Scanning Electron Microscopy and Energy-Dispersive X-ray Spectroscopy

Following treatment, the implants were carefully removed from their fixation blocks and prepared for scanning electron microscopy (SEM, FEI Quanta 450, FEI Deutschland GmbH, Frankfurt, Germany). For this purpose, samples were fixed with a solution containing 3 vol% glutaraldehyde and critical-point dried (EM CPD300, Leica, Germany) after dehydration in a graded ethanol series. Afterwards, samples were gold sputtered and images were taken by applying the Everhart–Thornley detector (ETD) back-scatter detector (BSE) combination. The surfaces of the implants were screened extensively and approximately 20 images were taken per sample. Representative images are shown. Material analysis was performed with energy-dispersive X-ray spectroscopy (EDX) [40].

## 3. Results

Prior to biofilm formation and decontamination treatment, regular implant surfaces as described by the manufacturer were present on the implants (Figure 2A). After biofilm formation, cells of the yeast *C. dubliniensis* (Figure 2B) and the Gram-positive bacterium *E. faecalis* (Figure 2C) were detectable using SEM.

### 3.1. Effect of Mechanical Debridement by Curettes

In a first approach, implants contaminated with *C. dubliniensis* (Figure 3A) and *E. faecalis* (Figure 3B) were mechanically debrided using metal curettes and treated with chlorhexidine. Both *C. dubliniensis* and *E. faecalis* were able to form biofilm-surviving chemo-mechanical debridement as indicated by the intact appearance of cells visible in SEM. In a parallel, independent set of experiments, similarly treated implants were not prepared for SEM but tested for bacterial growth and showed high numbers of colony-forming units [33].

The use of curettes led to visible alterations of the implant surface where the typical sandblasted large-grit acid-etched (SLA) surface was flattened (Figure 3A, middle part and Figure 3B, upper part of image), while the appearance of *C. albicans* and *E. faecalis* was unchanged, indicating unharmed cell structures.

Two explanations are possible for the flattened implant surface areas observed, e.g., deposition of material leading to leveling of the rough surface or destruction of the structure. An EDX scan of the altered surface showed the presence of titanium (Ti) as a main element of the flattened surface area in addition to the sputtered gold layer (Figure 4). As this exactly represents the material of the implant, deposition of unrelated material is very unlikely. In addition, it revealed traces of iron, most likely resulting from abrasion of the curette (Figure 4).

### 3.2. Effect of Air Particle Abrasion

Similarly, air particle abrasion combined with chlorhexidine irrigation was not able to inactivate *C. dubliniensis* and *E. faecalis* (Figure 5), deduced from the unchanged appearance of cells and from growth experiments [33]. Accumulations of crystals on the implant surface were clearly visible, and EDX scans support the idea that these were composed of sodium bicarbonate, which was the main ingredient in the air polishing powder used (Figure 6). Contrary to the use of curettes, air particle abrasion was not able to damage the implant surface or change surface roughness.

### 3.3. Effect of BDD Electrode Treatment

In contrast to the established treatment techniques applied above, electrochemical disinfection by BDD electrode application constituted the only treatment option able to securely inactivate microorganisms indicated by their altered shape (Figure 7), as described previously [41] and in independently carried out growth experiments [33]. More important for the purpose of this study, no change in surface topography was observed following disinfection. While lacking comparative statistical analysis, the SEM images obtained clearly show that the treatment modalities tested differed with respect to surface alterations of the implants considered.

## 4. Discussion

Given that ideal surfaces for dental implants promoting rapid osseointegration [42] but avoiding bacterial contamination are not yet available [4], peri-implantitis constitutes a clinical reality which has to be addressed. Current approaches in peri-implantitis treatment are aimed at disinfecting implant surfaces without altering manufacturer-generated surface features such as roughness. In an ideal situation, such an unaltered but cleaned surface would be conducive to the bone regeneration required for long-lasting osseointegration and hence for maintaining the high levels of oral-health-related quality of life achieved with initial implant-based treatment [43].

As derived from periodontal treatment, curettes have been used for mechanical debridement but, as shown here, the use of metal instruments harms the implant surface. As expected, the material used for the curette bears greater hardness as compared to titanium which does not lead to metal debris on the implant surface but to dislocation and alteration of the implant material. It seems to be impossible to mechanically debride an implant around its whole circumference and consequently bacteria do remain which cannot be removed by simple CHX irrigation. Air particle abrasion in this context seems even more critical as remnants of the powder material were shown and consequently, manufacturers do not recommend sodium bicarbonate for subgingival debridement but glycine-based materials. With both approaches, it seems to be impossible to reduce the surface roughness to a level which does not allow any biofilm attachment [44]. Additional polishing using rotary instruments can be anticipated to cause even greater contamination [45].

Other than potentially expected, the high level of hardness realized in BDD electrodes did not cause visible alterations of the implant surfaces. This may be due to the use of thin wires as electrodes requiring substantial treatment time for successful inactivation of microorganisms. It can be only speculated that enlarged electrodes, which would be suitable for delivering greater charge quantities in shorter periods of time, might be critical with respect to implant surface alterations. Also, here, bacteria remained on the implant surface but as they were inactivated, the patient’s immune system should be able to remove them in contrast to the living microorganisms left by the traditional treatment methods.

As with every in vitro study, several limitations have to be considered when interpreting the findings of this proof of principle study. Assuming that peri-implantitis would be treated by a surgical approach where the presence of saliva is generally seen as contamination, no attempt has been made to simulate the presence of, for example, artificial saliva. Similar to the use of air abrasion, the BDD electrodes could however also be applied without flap elevation where saliva might act as an additionally present electrolyte. With this study being designed as a proof of concept for the application of BDD electrodes for implant decontamination, no statistical analysis has been performed. Future studies have to be designed with a more quantitative approach and the results obtained here will serve for sample size calculation. While the present work showed that it was possible to predictably inactivate bacterial cells, the effect on human tissue is unknown and has to be evaluated prior to animal and clinical studies.

While being only in a prototypical development stage, BDD electrode arrays seem to be a promising treatment alternative for disinfecting dental implants in situ. Besides inactivating microorganisms, this treatment modality does not alter the implant surfaces thereby allowing for future bone regeneration [46].

## Figures and Tables

**Figure 1 materials-12-03977-f001:**
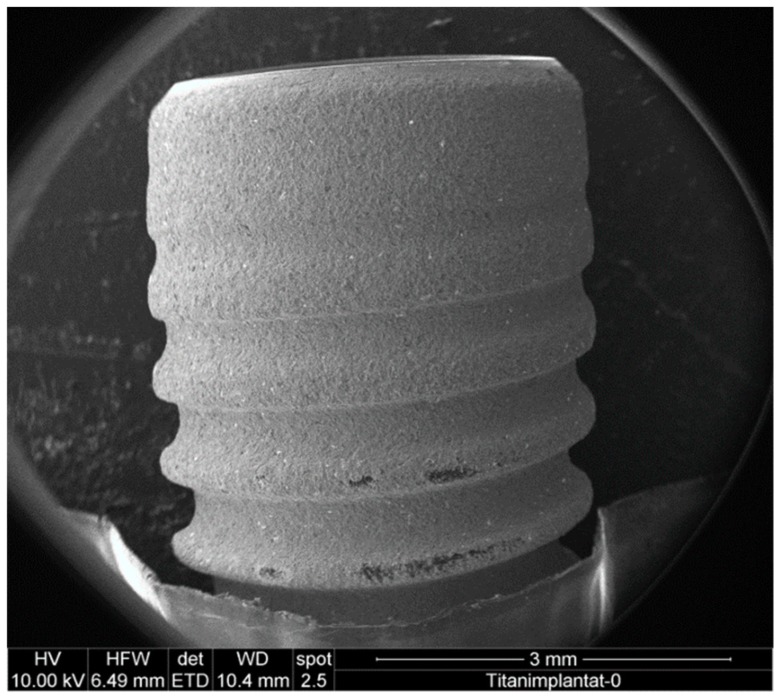
Low magnification image of a mounted implant. The sample is surrounded by aluminum tape to enhance connectivity and fix it on the sample plate.

**Figure 2 materials-12-03977-f002:**
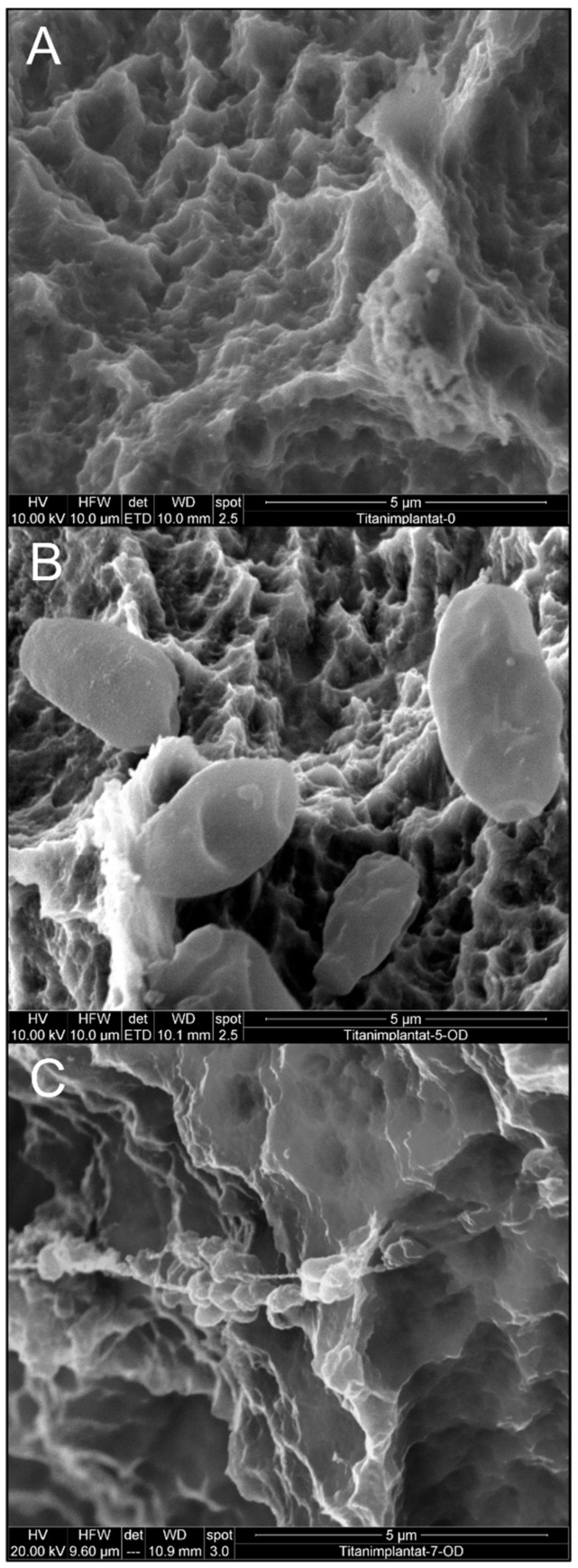
Scanning electron microscopy of implants before decontamination. Implant surface (**A**) before colonization, (**B**) with *C. dubliniensis* and (**C**) with *E. faecalis* biofilm.

**Figure 3 materials-12-03977-f003:**
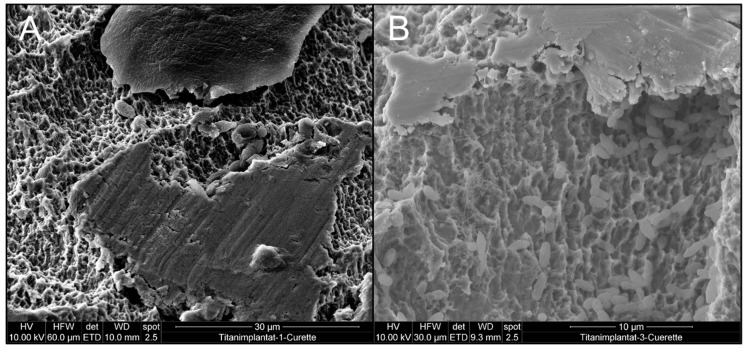
Effect of chemo-mechanical debridement on biofilm appearance and implant topography (**A**) implant colonized by *C. dubliniensis* and (**B**) *E. faecalis* biofilm.

**Figure 4 materials-12-03977-f004:**
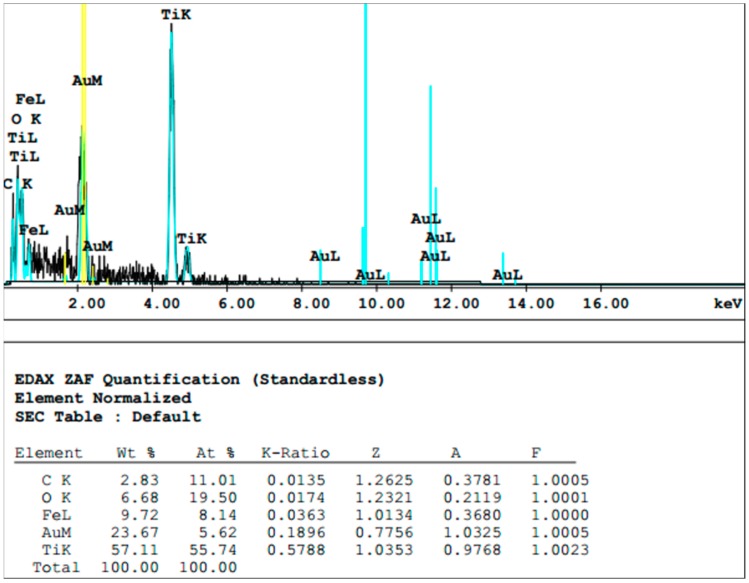
Element analysis of scratched implant surface. Ti and O were detected as the two main elements by energy-dispersive X-ray spectroscopy (EDX). In addition, gold used for sputtering was detected.

**Figure 5 materials-12-03977-f005:**
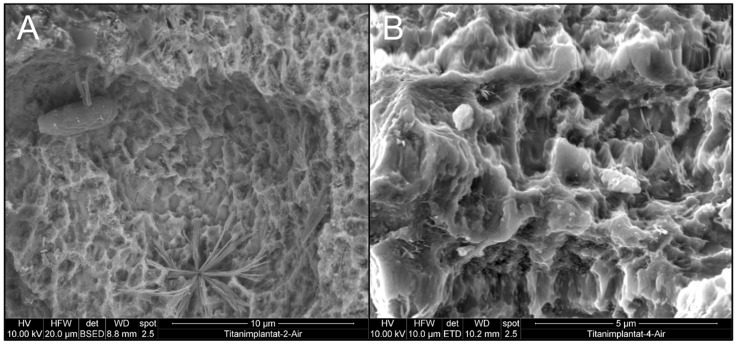
Effect of air abrasion and CHX irrigation on biofilm appearance and implant topography (**A**) implant colonized by *C. dubliniensis*, (**B**) *E. faecalis* biofilm. Please note the sodium bicarbonate crystal at the bottom of panel A.

**Figure 6 materials-12-03977-f006:**
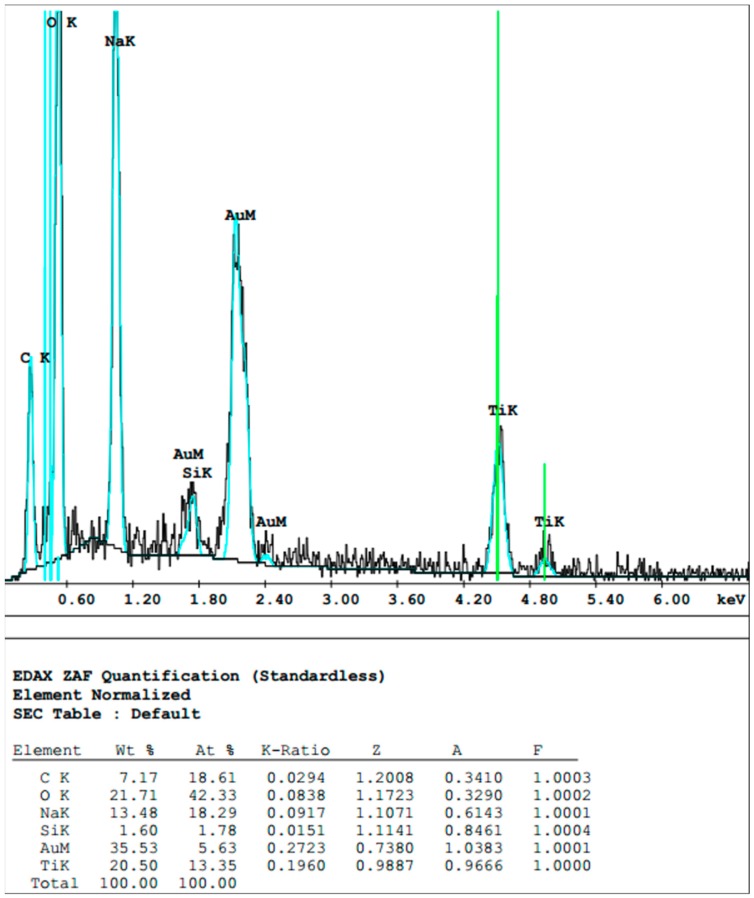
Element analysis of air polishing powder crystals on implant surfaces. C, O and N were detected as the three main elements of crystals by EDX besides titanium of the implant surface.

**Figure 7 materials-12-03977-f007:**
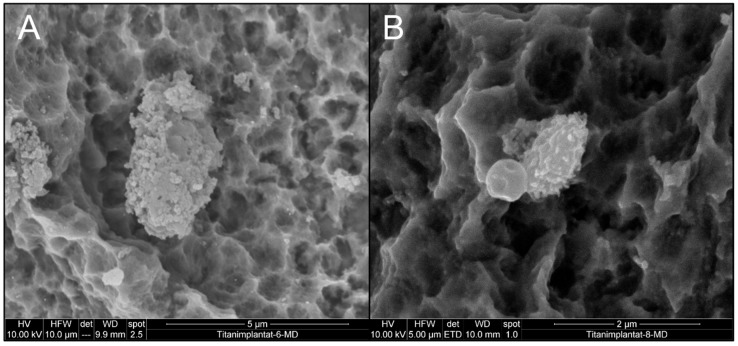
Effect of boron-doped diamond (BDD) electrode application on biofilm appearance and implant topography (**A**) implant colonized by *C. dubliniensis*, (**B**) *E. faecalis* biofilm. Please note the deteriorated surface of *C. dubliniensis* and *E. faecalis*.

**Table 1 materials-12-03977-t001:** Redox potential of selected reactive oxygen species.

Oxidation Product	Electrochemical Reaction	E Red (V)
OH•	HO• + H^+^ + e^−^ → H_2_O	2.80
O•	O• + 2H^+^ + 2e^−^ → H_2_O	2.42
O_3_	O_3_ + 2H^+^ + 2e^−^ → O_2_ + H_2_O	2.08
H_2_O_2_	H_2_O_2_ + 2H^+^ + 2e^−^ → 2H_2_O	1.76

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
