# Peer review of "Influence of In-Situ Electrochemical Oxidation on Implant Surface and Colonizing Microorganisms Evaluated by Scanning Electron Microscopy"

_materials, 2019, doi:10.3390/ma12233977_

Round 1

Reviewer 1 Report

The article is well written and the study is well conducted. The topic is interesting and focus the attention on the central theme of the implantology. 

However, I suggest to improve the discussion about the limits of the study:

it is not a clinical study so the effect in vivo should be detected in the future different type of microorganisms causes perimplantitis and the pathoetiology mechanisms are complex

Author Response

The article is well written and the study is well conducted. The topic is interesting and focus the attention on the central theme of the implantology. 

Re.: Thank you very much for the positive comments.

However, I suggest to improve the discussion about the limits of the study: it is not a clinical study so the effect in vivo should be detected in the future different type of microorganisms causes perimplantitis and the pathoetiology mechanisms are complex

Re.: We fully agree with the reviewer that several steps are required prior to implementing electrochemical disinfection in clinical practice. As already asked for by the other reviewers, we have added an introductory comment stating that this was a proof-of-principle study and we have added some aspects for future studies to the discussion section.

Reviewer 2 Report

The title is misleading. There is no peri-implantitis treatment. This study was about the effect of the in-situ electrochemical oxidation on the microbial-colonized implant surface under SEM.

To imitate the clinical situation, the artificial saliva should be used during the experiment.

The author should discuss the voltage and potential of using this approach in the animal model, human and chair-side situation.

Why were 2 specific strains of microbial organisms used in this study? Was the population of these microbial organism highest in periimplantitis microorganism population? Should the microbial of interest be Prevotella, Porphyromonas or Treponema.

It seems like chlorhexidine was incorporated in all the groups. What is the rationale to incorporate CHX in the experiment? Please discuss.

What was the conclusion on the bacterial inactivation based on?

What is the reproducibility rate? How many implants were subjected to each techniques?

What is the criteria to choose the area of SEM exposure?

Please discuss

Author Response

The title is misleading. There is no peri-implantitis treatment. This study was about the effect of the in-situ electrochemical oxidation on the microbial-colonized implant surface under SEM.

Re.: The title was changed as suggested and now reads “Influence of in-situ electrochemical oxidation on implant surface and colonizing microorganisms evaluated by scanning electron microscopy”

To imitate the clinical situation, the artificial saliva should be used during the experiment.

Re.: While the reviewer is correct that it is difficult to exclude saliva from a surgical field in the oral cavity, under ideal circumstances, a flap would be raised allowing for access to contaminated implant surfaces taking care that no secondary contamination occurs. In this context, saliva would have to be seen not as “reactive medium” but as a complicating factor. We have added this aspect to the discussion section where it reads “As with every in vitro study, several limitations have to be considered when interpreting the findings of this proof of principle study. Assuming that periimplantitis would be treated by a surgical approach where the presence of saliva is generally seen as contamination, no attempt has been made to simulate the presence of e.g. artificial saliva. Similar to the use of air abrasion, the BDD electrodes could however also be applied without flap elevation where saliva might act as an additionally present electrolyte.”

The author should discuss the voltage and potential of using this approach in the animal model, human and chair-side situation.

Re.: A corresponding text was added which reads “Recently, BDD electrodes have been described as being suitable for large-scale electrochemical disinfection of microbially contaminated water [28] and we were already able to show that also root canals and implants can be disinfected with low electric current ranging from 2.5 to 9 V which has to considered as being harmless for the patient”

Why were 2 specific strains of microbial organisms used in this study? Was the population of these microbial organism highest in periimplantitis microorganism population? Should the microbial of interest be Prevotella, Porphyromonas or Treponema.

Re.: We are aware that the genera mentioned are important members of dental biofilms. However, these microorganisms are either anaerobic or microaerophilic and, consequently, are even impaired by standard oxygen concentrations. For our experiments; we have chosen much more robust species. As suggested by the reviewer, this selection is better justified now by adding a paragraph which reads “As colonizing species, the yeast Candida dubliniensis and the Gram-positive bacterium Enterococcus faecalis were chosen. These pathogens were isolated from infected root canals by us recently [30,31], are frequently found in cases of peri-implantitis [32,33] and are in contrast to the anaerobic or microaerophilic members of the genera Prevotella, Porphyromonas and Treponema, which are highly oxygen-sensitive, fast growing and resistant against atmospheric oxygen concentrations.”

It seems like chlorhexidine was incorporated in all the groups. What is the rationale to incorporate CHX in the experiment? Please discuss.

Re.: CHX has indeed been used as rinsing solution and we have added a recent paper by Daubert and Weinstein as reference for its use.

What was the conclusion on the bacterial inactivation based on? What is the reproducibility rate? How many implants were subjected to each techniques?

Re.: After preparation of samples for SEM, all microorganisms and cells are dead and physiological tests with the samples are impossible. Therefore, the appearance of cells was taken as an indication of cell integrity and monitored (mentioned in l. 141-143, 149-150, 162), a method which was previously published by Mishra et al., 2017 (cited in the text). In parallel to this SEM-based study, microbiological experiments were carried out, which are mentioned, referenced and discussed now (Reference # 33). Tests of growth behavior of bacteria and yeasts was highly reproducible with at least three independent biological replicates per species and treatment method applied (Reference # 33).

What is the criteria to choose the area of SEM exposure?

Re.: We have added the following explanation to the respective section “The surface of implants was screened extensively and approximately 20 images were taken per sample.”

Reviewer 3 Report

Dear Authors,

After the review process, I have several comments: you should insert references in all Materials and Methods sections; you should insert a statistical section or to detail why it is not necessary; you should present a limitation of the study; you should insert possible interactions with other dental materials.

Best regards!

Author Response

After the review process, I have several comments: you should insert references in all Materials and Methods sections;

Re.: As suggested, we added several new references and in addition improved the explanation of techniques applied.

you should insert a statistical section or to detail why it is not necessary;

Re.: As per reviewer #2 suggestion, we have added sample sizes as an indicator for reproducibility. This study was intended as a proof of concept evaluation for electrochemical disinfection and we have added that additional work with sophisticated model situations and much greater sample size is required including statistical analysis. This was added as a limitation of the study where it now reads “With this study being designed as a proof of concept for the application of BDD electrodes for implant decontamination, no statistical analysis has been performed. Future studies have to be designed with a more quantitative approach and the results obtained here will serve for sample size calculation.”.

you should present a limitation of the study;

Re.: Please see comment above

you should insert possible interactions with other dental materials.

Re.: We are currently working on evaluating the interactions with host cells i.e. does electrochemical disinfection cause harm to the patient’s tissue and this aspect was added to the discussion section where it now reads “While the present work showed that it was possible to predictably inactivate bacterial cells, the effect on human tissue is unknown and has to be evaluated prior to animal and clinical studies.”

Reviewer 4 Report

See attached

Author Response

Manuscript compares the novel method of electrochemical disinfection to mechanical debridement and air particle abrasion for treatment of peri-implantitis. As the authors explain, these traditional methods may impair the surface topography such that bacterial recolonization is favoured or re-ossointegration is obstructed. Electrochemical oxidation using BDD electrodes is one of a number of potential novel routes. An in vitro study considering pathogens commonly found in cases of peri-implantitis has been carried out with a focus on the influence on surface topography.

The work appears novel and of clear scientific value and societal impact. The design of the in vitro experiment also appears well-thought out and representative of the clinical scenario. The paper is well written, adequately referenced and discussions are generally sound. However as I will highlight I have some concerns regarding the use of only a single analytical tool, scanning electron microscopy. As a consequence of this I also have some concerns as to the accuracy of the conclusions drawn.

Re.: Thank you for the positive comments and helpful suggestions.

Specific concerns with results and discussion:

Appearance of cells in SEM is given as evidence of biofilm survival – surely some analysis of cell activity should also have been considered, this is particularly true in the case of the BDD treatment where cell appearance is the only evidence provided for inactivation of microorganisms. Additionally only 1 or 2 cells are imaged.

It is mentioned that growth experiments were carried out independently in the case of the BDD treatment, this data is in my mind essential to the paper as well as growth experiments for the other treatment routes.

Re.: After preparation of samples for SEM, all microorganisms and cells are dead and physiological tests with the samples are impossible. Therefore, the appearance of cells was taken as an indication of cell integrity and monitored (mentioned in l. 141-143, 149-150) a method which was previously published by Mishra et al., 2017 (cited in the text). In parallel to this SEM-based study, microbiological experiments were carried out, which are mentioned, referenced and discussed now (Reference #33).

            Tests of growth behavior of bacteria and yeasts was highly reproducible with at least three independent biological replicates per species and treatment method applied (Reference #33).

Figure 2 – the suggestion is made that chemomechanical debridement results in destruction of the sandblasted structure, with the implication being it is worn away. The micrographs of figure 2 however appear to show an additional layer on top of the sand blasted topography – the authors should discuss this.

Re.: Thank you for this valuable hint. We had the same suspicion and therefore carried out the EDX analyses, which clearly demonstrated that the altered surface consists of titanium. As suggested, this result is better presented and discussed now where it reads “Two explanations are possible for the flattened implant surface aereas observed: deposition of material leading to leveling of the rough surface or destruction of the structure. An EDX scan of the altered surface showed the presence of titanium (Ti) as a main element of the flattened surface area in addition to the sputtered gold layer (Figure 4). As this exactly represents the material of the implant, deposition of unrelated material is very unlikely. In addition, it revealed traces of iron, most likely resulting from abrasion of the curette (Figure 4).”

Was any calibration of the EDX carried out? Discussion of presence of C,O,N analysed with EDX should be discussed only with extreme caution. Using method for analysis of any iron presence is however much more sensible.

Re.: Our EDX was not calibrated for each element. Based on the valuable hint of the reviewer, discussion of results was changed as suggested (see comment above and paragraph 3.2 Effect of air particle abrasion).

In conclusion:

Whilst the paper does indeed show that BDD treatment does not damage surface topography not enough evidence is included to support that it is successful in disinfecting the implants.

Re.: As mentioned above, this study focusses on surface alterations and material aspects. Inactivation of microorganisms can only be suggested by SEM, but not proven. This is part of a different manuscript cited in the text [33].

Reviewer 5 Report

This is an interesting study on the effect of different super-face treatments on implants in morphology and bacterial colonization ability.

Some criticisms are present:

-Line 18: in the abstract an initial sentence on peri-implantitis should be inserted

-Lines 24-27 too specific statements about the treatments performed must be removed from the abstract

-Line 36 replace boron-doped diamond with periimplantitis

--Line 40 The introduction should be modified. First of all, some initial considerations on the possible causes of periimplantitis and on the possible microorganisms involved

-Line 44 Some important considerations must be added on the possible effect of disinfection treatments of implants on their biocompatibility. In this regard, I advise you to add the following scientific work in refeerence that can help reflection:

- Chieruzzi M., Pagano S.,  Lombardo G.,  Marinucci L.,  Kenny J.M.,  Torre L.,  Cianetti, S.

Effect of nanohydroxyapatite, antibiotic, and mucosal defensive agent on the mechanical and thermal properties of glass ionomer cements for special needs patients

Journal of Materials Research 2018, 33(6); 638-649.

-Line 68 At the end of the Introduction the null hypotheses of the study must be inserted, in this case for example that the different surface treatments do not show differences on the implant surface

-Figure 1 Measurement scales are not visible. modify them

Figures 2,3,4 I would recommend putting the implant surface without any treatment in each figure in order to make a comparison

Author Response

This is an interesting study on the effect of different super-face treatments on implants in morphology and bacterial colonization ability.

Re.: Thank you for the positive comment.

Some criticisms are present:

-Line 18: in the abstract an initial sentence on peri-implantitis should be inserted

Re.: “Peri-implantitis is a worldwide increasing health problem, caused by infection of tissue and bone around an implant by biofilm-forming microorganisms” was added as an introductory sentence

-Lines 24-27 too specific statements about the treatments performed must be removed from the abstract

Re.: Removed as suggested.

-Line 36 replace boron-doped diamond with periimplantitis

Re.: In this case, we did not follow the reviewer’s advice, since the sentence would be incorrect with the suggested change.

--Line 40 The introduction should be modified. First of all, some initial considerations on the possible causes of periimplantitis and on the possible microorganisms involved

Re.: Trying to stay focused, we have merely added one sentence on the pathogenesis of periimplantitis which reads “Besides patient-related factors such as preceeding periodontitis and general health, restorative factors such as the design of the superstructure and hygiene efforts are currently understood as playing a role in the multifactiorial pathogenesis of periimplantits”

-Line 44 Some important considerations must be added on the possible effect of disinfection treatments of implants on their biocompatibility. In this regard, I advise you to add the following scientific work in refeerence that can help reflection:

Chieruzzi M., Pagano S.,  Lombardo G.,  Marinucci L.,  Kenny J.M.,  Torre L.,  Cianetti, S.

Effect of nanohydroxyapatite, antibiotic, and mucosal defensive agent on the mechanical and thermal properties of glass ionomer cements for special needs patients Journal of Materials Research 2018, 33(6); 638-649.

Re.: Thank you very much for pointing us to this paper which we have added to the introduction where it now reads “While it has so far not been managed, to create implant surfaces which are conducive to osseointegration but avoid bacterial colonization as is performed for restorative materials”

-Line 68 At the end of the Introduction the null hypotheses of the study must be inserted, in this case for example that the different surface treatments do not show differences on the implant surface

Re.: Given that this investigation was intended as a proof of principle study using merely descriptive comparisons, we have been cautious in formulating a null hypothesis which could then be rejected based on comparative statistical analysis. We have now added a somewhat weaker statement to the Introduction and stated in the results section that based on the SEM images obtained a clear difference between treatment modalities with respect to implant surface alterations could be seen.

Figure 1 Measurement scales are not visible. modify them

Re.: The figure was enlarged to improve the visibility of scale bars.

Figures 2,3,4 I would recommend putting the implant surface without any treatment in each figure in order to make a comparison

Re.: The untreated implant surface was already shown in figure 1. We did not add additional images to figures 2 to 4 to avoid reduction of the size of treated implants (see valuable comment to Fig. 1).

Reviewer 6 Report

Dear Authors,

Your manuscript is really interesting.

I think that it could be modified from communication to "Article".

In keyword section please use medical subject headings words.

Please subdivide intro section into two subparagraph, with an aim subparagraph.

On M&M section I suggest to add a scheme or a optical microscope pic of the dental implant.

I suggest two recent publication with additional information for Your discussion section: 

1. Cicciu, M.; Fiorillo, L.; Herford, A.S.; Crimi, S.; Bianchi, A.; D'Amico, C.; Laino, L.; Cervino, G. Bioactive Titanium Surfaces: Interactions of Eukaryotic and Prokaryotic Cells of Nano Devices Applied to Dental Practice. Biomedicines 2019, 7, doi:10.3390/biomedicines7010012.

2. Cervino, G.; Fiorillo, L.; Iannello, G.; Santonocito, D.; Risitano, G.; Cicciù, M. Sandblasted and Acid Etched Titanium Dental Implant Surfaces Systematic Review and Confocal Microscopy Evaluation. Materials 2019, 12, 1763, doi:https://doi.org/10.3390/ma12111763.

It is essential to cite how implant rehabilitation are correlated with QoL.

Bramanti, E.; Matacena, G.; Cecchetti, F.; Arcuri, C.; Cicciù, M. Oral health-related quality of life in partially edentulous patients before and after implant therapy: A 2-year longitudinal study. 2013, 6, 37-42.

You did not provide a conclusion section, please add some future perspective at the end of Your manuscript

Thank You

Author Response

Your manuscript is really interesting.

Re.: Thank you for the positive comment.

I think that it could be modified from communication to "Article".

Re.: Done.

In keyword section please use medical subject headings words.

Re.: Keywords have bene changed to “Candida; Enterococcus; debridement; peri-implantitis; disinfection; microscopy, electron, scanning”

Please subdivide intro section into two subparagraph, with an aim subparagraph.

Re.: As already asked for by Reviewer #5, we have added a paragraph stating the aim of this proof of principle study.

On M&M section I suggest to add a scheme or a optical microscope pic of the dental implant.

Re.: A low magnification image of a mounted implant was added as new Figure 1.

I suggest two recent publication with additional information for Your discussion section:

Cicciu, M.; Fiorillo, L.; Herford, A.S.; Crimi, S.; Bianchi, A.; D'Amico, C.; Laino, L.; Cervino, G. Bioactive Titanium Surfaces: Interactions of Eukaryotic and Prokaryotic Cells of Nano Devices Applied to Dental Practice. Biomedicines 2019, 7, doi:10.3390/biomedicines7010012. Cervino, G.; Fiorillo, L.; Iannello, G.; Santonocito, D.; Risitano, G.; Cicciù, M. Sandblasted and Acid Etched Titanium Dental Implant Surfaces Systematic Review and Confocal Microscopy Evaluation. Materials 2019, 12, 1763, doi:https://doi.org/10.3390/ma12111763.

Re.: Thank you very much for pointing us to these two very important references, which we have now included in our manuscript

It is essential to cite how implant rehabilitation are correlated with QoL.

Bramanti, E.; Matacena, G.; Cecchetti, F.; Arcuri, C.; Cicciù, M. Oral health-related quality of life in partially edentulous patients before and after implant therapy: A 2-year longitudinal study. 2013, 6, 37-42.

Re.: While it is generally accepted that dental implant treatment bears benefits for our patients, this paper gives real evidence and hence we have included it in our list of references and discussed its implications on periimplantitis treatment.

You did not provide a conclusion section, please add some future perspective at the end of Your manuscript

Re.: As already asked for by Reviewer #3, we have added a section on future studies to be performed prior to introducing electrochemical disinfection into clinical practice.

Round 2

Reviewer 2 Report

The avoidance of selection bias of SEM images could be clarified to improve the quality of the manuscript. 

Reviewer 4 Report

All comments to the authors have been adequately addressed.

Reviewer 5 Report

All suggestions was done. I suggest acceptation of paper

Reviewer 6 Report

Dear Authors, thank You for agreeing my notes.

Now Your manuscript is suitable for publication

Thank You

Kind regards